# Enhancing the Antimicrobial Effect of Ozone with *Mentha piperita* Essential Oil

**DOI:** 10.3390/molecules28052032

**Published:** 2023-02-21

**Authors:** Alin-Daniel Floare, Ramona Dumitrescu, Vlad Tiberiu Alexa, Octavia Balean, Camelia Szuhanek, Diana Obistioiu, Ileana Cocan, Alina-Georgeta Neacsu, Iuliana Popescu, Aurora Doris Fratila, Atena Galuscan

**Affiliations:** 1Translational and Experimental Clinical Research Centre in Oral Health, University of Medicine and Pharmacy “Victor Babes”, Eftimie Murgu Sq. no 2, 300041 Timisoara, Romania; 2Department of Preventive, Community Dentistry and Oral Health, University of Medicine and Pharmacy “Victor Babes”, Eftimie Murgu Sq. no 2, 300041 Timisoara, Romania; 3Orthodontic Research Center (ORTHO-CENTER), Faculty of Dental Medicine, “Victor Babes” University of Medicine and Pharmacy, Eftimie Murgu Sq. No 2, 300041 Timisoara, Romania; 4Faculty of Veterinary Medicine, University of Life Sciences “King Mihai I” from Timisoara, Calea Aradului No. 119, 300641 Timisoara, Romania; 5Faculty of Food Engineering, University of Life Sciences “King Mihai I” from Timisoara, Calea Aradului No. 119, 300645 Timisoara, Romania; 6Faculty of Agriculture, University of Life Sciences “King Mihai I” from Timisoara, Calea Aradului No. 119, 300641 Timisoara, Romania; 7Faculty of Dental Medicine, Ludwig-Maximilian University Munich, Goethestr. 70, 80336 Munich, Germany

**Keywords:** *Mentha piperita*, essential oil, GC-MS, ozonation, gram-positive, gram-negative bacteria

## Abstract

This study aimed to obtain and analyse *Mentha piperita* essential oil (MpEO) for the prospect of being used as an enhancement agent for the antimicrobial potential of ozone against gram-positive and gram-negative bacteria and fungi. The research was done for different exposure times, and it gained time–dose relationships and time–effect correlations. *Mentha piperita* (Mp) essential oil (MpEO) was obtained via hydrodistillation and further analysed by using GC-MS. The broth microdilution assay was used to determine the strain inhibition/strain mass growth by using spectrophotometric optical density reading (OD). The bacterial/mycelium growth rates (BGR/MGR) and the bacterial/mycelium inhibition rates (BIR/MIR) after ozone treatment in the presence and absence of MpEO on the ATTC strains were calculated; the minimum inhibition concentration (MIC) and statistical interpretations of the time–dose relationship and specific *t*-test correlations were determined. The effect of ozone on the following tested strains at maximum efficiency was observed after 55 s of single ozone exposure, in order of effect strength: *S. aureus* > *P. aeruginosa* > *E. coli* > *C. albicans* > *S. mutans*. For ozone with the addition of 2% MpEO (MIC), maximum efficacy was recorded at 5 s for these strains, in order of effect strength: *C. albicans* > *E. coli* > *P. aeruginosa* > *S. aureus* > *S. mutans*. The results suggest a new development and affinity regarding the cell membrane of the different microorganisms tested. In conclusion, the use of ozone, combined with MpEO, is sustained as an alternative therapy in plaque biofilm and suggested as helpful in controlling oral disease-causing microorganisms in medicine.

## 1. Introduction

Ozone is a chemical compound consisting of three oxygen atoms (O_3_, triatomic oxygen) in an energy form larger than normal atmospheric oxygen (O_2_). It is one of the most potent oxidants with a high oxidation capacity [1] and also with bactericidal, fungicidal, and virus-inactivating effects. Ozone is well-known for its use in water purification and drinking water treatment as well as for being used in heavily infected wounds such as burns, diabetic feet, and ulcus cruris, lesions infected with germs resistant to most antibiotics, such as MRSA (e.g., methicillin-resistant *Staphylococcus aureus*) [2].

Ozone allows microbial flora to pass from acidogenic and acidic microorganisms to normal oral commensals. In addition, ozone has a significant environmental advantage represented by rapid degradation, a process that clinically causes low cytotoxicity after contact with organic compounds. All of these features suggest that ozone could be widely used soon in restorative and preventive dentistry [1,2,3,4,5].

Due to its high oxidation power, ozone can oxidise the bacterial cell wall, causing its lysis, and the pyruvic acid produced by bacteria can then turn it into acetic acid and carbon dioxide.

In recent years, in the search for new natural therapeutic solutions, the use of essential oils (EOs) in alternative medicine and dentistry represents modern trends with a positive impact on the patient [6,7]. *Mentha piperitha* essential oil (MpEO) has multiple uses in medicine and dentistry. In dentistry, it was used as part of a mouthwash formula, positively influencing the management of oral mucositis. It was demonstrated that the MpEO, when loaded in chitosan polymeric nanogel, was influential in inhibiting *Streptococcus mutans*, an important pathogen of dental caries in humans. In addition, the MpEO-nanogel inhibited the formation of biofilms on the dental surface [8]. In particular, the antimicrobial effect of MpEO was studied against gram-positive and gram-negative bacteria [9,10]. The combination of conventional therapy with natural compounds such as EOs leads to synergistic/antagonistic effects. The interaction of MpEO with amphotericin B as a conventional antimicrobial agent revealed an antagonistic effect against *Candida albicans* and a synergistic effect against *Escherichia coli* and *Candida albicans*. However, moderate influence against *Staphylococcus aureus* was observed [11]. The inhibitory effect of MpEO against multidrug-resistant clinical pathogenic bacteria, including *Streptococcus pyogenes*, *Enterococcus faecalis*, and methicillin-resistant *Staphylococcus aureus*, was also reported [12].

The use of EOs as antimicrobial agents in water disinfection was recently studied. EOs of ginger, turmeric, lavender, and tulsi enhanced the disinfection rate up to five times when an innovative hybrid hydrodynamic cavitation process was applied [13]. MpEO’s efficiency against *Pseudomonas aeruginosa* and *Staphylococcus aureus* using two different cavitating reactors and hydrodynamic cavitation was higher than 99%. The advantage of these procedures refers to enhancing the disinfection rate concurrently with significant lowering of the oil dose [14].

The use of ozonated vegetal oils was previously evaluated in terms of their antibacterial and antifungal activity with application in skin treatments [15,16,17] and dentistry [18,19,20]. In oral medicine, the control of periodontal and infectious diseases with oral microorganisms based on ozonated oils, such as sunflower, olive, and groundnut, were reported [20,21]. In addition, commercial formulations, such as ozone gel with olive oil in its composition, were patented to control the formation of dental plaque [22,23]. Indeed, ozonated sunflower oil (Oleozon^®^) was tested against pathogens such as *S. aureus*, *Enterococcus faecalis*, *S. pyogenes*, *Escherichia coli*, *Pseudomonas aeruginosa*, and different species of *Mycobacterium T.*, and the results confirmed the antimicrobial potential of this natural formulation. The MICs varied between 2.37 and 9.95 mg/mL for mycobacteria and between 1.18 and 9.5 mg/mL for all other bacteria [24].

Even if studies in the literature indicate a wide use of ozonated vegetable oils in medicine, the possibility of ozonation of essential oils from medicinal plants, or the association of the two elements, to our knowledge, was only briefly reported in the specialised literature. Some studies have been carried out on the influence of ozonation on the chemical composition of essential oils, emphasising that the presence of ozone does not change their chemical composition [25,26]. In addition, some commercial products are based on vegetal oils with essential oils added, but they do not present scientific support for antimicrobial effects [22].

This study aims to analyse the antimicrobial potential of ozone alone and ozone enhanced with MpEO against the following organisms at different times of exposure: bacteria *Escherichia coli* (*E. coli*), *Staphylococcus aureus* (*S. aureus*), *Pseudomonas aeruginosa* (*P. aeruginosa*), *Streptococcus mutans* (*S. mutans*), and fungi *Candida albicans* (*C. albicans*). In this regard, two indicators, the bacterial/mycelium growth rate (BGR/MGR) and the bacterial/mycelium inhibition rate (BIR/MIR), were calculated after ozone treatment in the presence and absence of MpEO on the ATTC strains, and the minimum inhibition concentrations (MIC) of ozone in single and combined forms were determined.

To our knowledge, this is the first study that intends to highlight the antagonistic/synergistic effect obtained by combining ozone with MpEO against pathogenic bacteria and fungi.

## 2. Results

### 2.1. The Obtaining and Characterisation of EOs

The chemical composition of MpEO is presented in Table 1.

The extraction yield of MpEO from the aerial parts of *Mentha piperita* prelevated was small (3.50%). In MpEO, 95.169% of terpene compounds in percentages over 1% were identified, of which the majority were oxygenated monoterpenes (83.967%), followed by monoterpene hydrocarbonates (5.792%) and sesquiterpene hydrocarbonates (5.410%). The main chemical compounds found in MpEO were menthol (32.658%), menthyl acetate (23.643%), and menthone (10.322%).

### 2.2. Evaluation of MpEO Antimicrobial Activity and Ozone Potential as Single and Enhanced with MpEO Antimicrobial Agent

Figure 1, Figure 2, Figure 3, Figure 4, Figure 5 and Figure 6 show the bacterial inhibition rates (BIR%)/mycelial inhibition rate (MIR%) against analysed strains of MpEO, ozone, and MpEO after ozone application, which were calculated according to Formula (3), and the bacterial growth rates in relation to the time of treatment application, which was calculated according to Formula (2).

Concerning MpEO’s efficacy against the strains tested, the results are presented in Figure 1. The results suggest that the efficacy of MpEO starts at the first concentration tested, meaning that the minimum concentration tested was 2%. *S. mutans* and *S. aureus* achieved a mean efficacy of 30% BIR at the first concentration tested, with their maximum efficacies reaching 50.13% and 67.59% BIR, respectively. Regarding *P. aeruginosa*, antibacterial activity evolved with a positive correlation to the concentration tested, with BIR% values ranging from 48.32% to 67.59%. The antimicrobial activity against *E. coli* was also recorded with BIR% values ranging from 62.08% to 74.06%. *C. albicans* was the only strain tested that showed a negative correlation; efficacy decreased alongside the increase in concentration. The BIR% value recorded for 2 µL was 67.54%, and for 32 µL, the BIR% was 51.07%, a decrease in value by almost 25%. However, even if the trend was negative, MIC was reached at the 2% concentration tested.

Comparing the BIR percentages concerning the ozone effect on *S. mutans*, the first effective duration was 50 s, with a value of 7.77% compared to the control (Figure 2). All of the values obtained in the previous tests were negative. The evolution trend was positively correlated with the duration of ozonation. The maximum duration tested, 240 s, reached a BIR% of 24.13% compared to the negative control, which was 0%. The negative values of BIR showed synergistic activity of ozone with the *S. mutans* strain tested; the values obtained proved a strain-boosting effect demonstrated by a bacterial mass growth of 2% compared to the positive control in the duration interval between 5 s and 45 s, with values ranging from −16.24% to −0.18% (Figure 2a).

All of the results changed when the ozone activity was enhanced by MpEO and used at the lowest MIC (2%). As a result, the first effective duration became half of the duration needed by ozone alone, 25 s, with a BIR% of 8.33%. Even if the starting point was similar in the first duration tested (16.24% compared to 16.97%), evolution improved compared to the inhibition caused by ozone alone. All of the subsequent durations used in the research showed higher BIR% than any initial durations, proving a BIR% enhancement rate of 15% on average. The inhibitory effects of the ozone and ozone enhanced with *M. piperitha* treatments were also revealed from the bacterial growth rate–time relationship, as shown in Figure 2b. Bacterial growth tended to decrease with ozone treatment time. The time–inhibition relationship clearly shows the inhibiting effect against *S. mutans* when ozone is applied in association with MpEO compared to simple ozone, which justifies this therapeutic approach.

Figure 3a summarises the ozone’s antimicrobial activity on *S. aureus*, both alone and enhanced with MpEO. Ozone was effective against *S. aureus* with BIR values between −14.75% and 80.91%, obtained from all of the periods tested. Concerning the growth-boosting effect, it was present from 5 s to 45 s, just as for *S. mutans*, and the inhibitory values were obtained after 50 s. These results are not superior to the ones recorded in the test. Positive BIR values started at 50 s with a value of 4.73%; the value registered as the MIC and increased up to 80.91%, the value reached after 240 s. The BIR values after 55 s were higher against the *S. aureus* and *S. mutans* strains.

Regarding the antimicrobial effect on *S. mutans*, the MIC was reached at 25 s; concerning *S. aureus*, inhibition was proven at the first duration tested, 5 s, with a BIR% value of 37.61%. All of the subsequent durations tested showed higher inhibition efficacy for ozone enhanced by MpEO, with an average enhancement rate of 65%.

The time–bacterial growth relationship demonstrates the inhibitory effect against *S. aureus* (Figure 3b). A decrease in bacterial growth over time can be observed both in the treatment with ozone only and in the treatment with ozone and MpEO, but higher bacterial inhibition can be noticed in the combined application of ozone with MpEO.

Concerning the inhibition of *P. aeruginosa*, ozone tested alone proved less effective until after 120 s, when the MIC for *P. aeruginosa* was recorded at a BIR value of 3.54%. All of the data were collected between 5–60 s, proving a positive strain-boosting effect. The effects correlated with the duration of the test, and negative inhibitory values are presented in Figure 4a.

Regarding ozone/MpEO efficacy, the effect was similar to the activity against *S. mutans*. MIC proved to be effective at 5 s with a difference from the ozone alone of 56.66%. The trend was evident, as inhibitory efficacy grew over time. The ozone alone reached the MIC at 120 s, the ozone enhanced with MpEO demonstrated the MIC at 5 s, and the difference at 120 s was a 67.73% improvement of BIR%. The decreasing trend of bacterial development over time was also observed, as shown in Figure 4b; it is worth mentioning that the growth rate was lower when supplementing the treatment with MpEO. Bacterial growth rate BGR (%) with the use of ozone–MpEO was lower (between 22–60%) than when the treatment was done only with ozone (between 78–120%) (Figure 4b).

Figure 5a presents the results regarding the 15 time durations tested on *E. coli*. The results show a positive antimicrobial effect recorded against *E. coli* (with the BIR% ranging from 0.16% to 39.07%) for the ozone-alone treatment tested, starting from 10 s.

The only negative value was presented at 5 s. Even if time had a relatively weak influence on the value of BIR recorded, with 0.64% at 15 s and 4.05% at 55 s, the effect was still inhibitory considering that the value obtained was reported to be 0% growth of the control strain. The last four durations tested proved good inhibitory results ranging from 11.39% to 39.07% BIR. The values obtained for the ozone enhanced with MpEO show the inhibitory pattern improved over what was presented in the previous test. For ozone alone, the BIR% was −0.19, and for ozone enhanced with MpEO, the BIR% was 61.45%. The values obtained present an improvement of BIR% after MpEO enhancement by approximately 70%. Figure 5b shows the decrease of bacterial growth over time, especially for ozone treatment enhancement with MpEO.

Regarding the inhibition of *C. albicans*, there was a positive correlation between the antimicrobial effect with ozonation duration, as seen in Figure 6a. The 120 s test was the first active, with a MIR of 1.6%, while the final test at 240 s showed a mycelial inhibitory percentage of 25.19%.

The values obtained for the enhanced ozone with MpEO show a clear improvement in results. For the single ozone tested at 5 s, the MIR% was −29.84%, and for the ozone enhanced with MpEO, the value reached 72.89%; all of the other durations tested confirmed an average rise in efficacy by 50%. The inhibitory activity against *C. albicans* was also demonstrated by BGR (%), as presented in Figure 6b. The bacterial growth rate was higher for the use of ozone alone (between 78–130%) than when the treatment of ozone and MpEO was applied (20%).

Figure 7 and Figure 8 present the inhibition percentages of all of the strains tested at different periods selected in the research. The difference in the figures regarding the strains and times of exposure to ozone is due to the fact that the data presented in Figure 8 are complementary to the data presented in Figure 7. The values presented in Figure 8 are results obtained from the first part of the analysis, from which we selected only the times necessary for obtaining the MIC for each strain, and all of the extra periods of time outside of the necessary timeframe were untested.

Concerning the ozone tested as a single element, the best results were on *S. aureus*, with the inhibition value reaching 80.91% at 240 s. The second strain most affected by ozone alone was *P. aeruginosa*, with a maximum inhibitory response of 54.21%, followed by *E. coli* at 39.07%. *S. mutans* and *C. albicans* showed similar values at maximum exposure: 24.13% and 25.18%, respectively. The results suggest the best effect was recorded for *C. albicans* with BIR% up to 80.71%, followed by *E. coli* (73.38%), *P. aeruginosa* (71.24%), *S. aureus* (69.54%), and finally, *S. mutans* (36.52%).

Regarding the BIR/MIR (%) values of MpEO (Figure 8), all bacterial strains were inhibited. For each strain, the trend was positive, with BIR% values ranging from 36.59% to 74.06%. There was a difference for *C. albicans,* where inhibition was present with MIR% values ranging from 67.54% to 51.07%, the difference being made by the negative trend of inhibition; the inhibitory values decreased with increases in EO concentration. For *C. albicans*, the increase in MpEO concentration negatively correlated with inhibition values.

The minimum inhibition concentrations (MIC) tested on the ATTC strains is presented in Table 2. MIC was determined as being the lowest time expressed in seconds of ozone exposure that inhibited the visible growth of cells.

Following statistical analysis with a *t*-test, the values obtained are recorded in the table presented as a Appendix A, wherein significant differences (*p* < 0.05) are recorded between the ozone-treated and ozone-MpEO-treated samples.

## 3. Discussion

### 3.1. The Obtaining and Characterisation of EOs

The extraction yield of MpEO from the aerial parts of *Mentha piperita* prelevated from spontaneous flora was small (3.50%) but similar to the results reported in the literature (3.22%) [27].

Similar compositions have been reported in other studies regarding the chemotypes identified in MpEOs. Oxygenated monoterpenes predominated: 60.826% in MpEOs obtained from the aerial parts of *Mentha piperita* originating from the spontaneous flora of Romania [28], and 73.9–94.8% obtained from Mentha species from China [29]. Yield rates for menthol include the following: 36.02% [30], 45.05% [31], 32.14% [32], 10.6–66.6% (depending on cultivar and harvest stage), and 11.57% [33]; for menthone, 20.14% [34] and 5–35% [35] were identified in the majority of EOs originating from *Mentha piperita* species cultivated in different geographical areas.

### 3.2. The Ozone Potential as Single and Enhanced with MpEO Antimicrobial Agent

The results obtained regarding the antimicrobial effect of ozone on the tested strains show that, in terms of the effect related to the effective exposure time, it decreased in the following order: *E. coli* > *S. aureus* = *S. mutans* > *P. aeruginosa* = *C. albicans*. Regarding the intensity of the effect of ozone on the tested strains, maximum efficiencies were observed, after 240 s of ozone exposure, in decreasing order of strength: *S. aureus* > *P. aeruginosa* > *E. coli* > *C. albicans* > *S. mutans*.

All of the results obtained using the OD values and, subsequently, the BIR values demonstrate the synergistic effect of the ozone and MpEO mixture. This affirmation is sustained by the fact that the inhibitory rates obtained in the mixture increased compared to the inhibitory rates obtained for ozone and MpEO tested as single compounds. This fact is possibly a result of a dual effect: the oxidation reaction of the bacterial wall produced by the ozone, potentiated with the MpEO mechanism of weakening the cell walls of resistant bacteria, damages or kills these cells.

The effects of ozone on various bacterial and fungal strains and their mechanisms of action were previously reported in specialised studies [36]. Giuliani et al., 2018 explained that ozone attacks numerous cellular constituents such as proteins, unsaturated lipids, enzymes, and nucleic acids in the cytoplasm [36]. When ozone interacts with the cell wall, an oxidative burst occurs, creating a tiny hole in the cell wall; thus, the bacterium loses its shape, and after multiple collisions with ozone, the cell dies in only a few seconds [36]. Another explanation regarding the antimicrobial activity was given by Srinivasan SR, Amaechi BT, which highlighted the solid oxidising potential of ozone that degrades the cytoplasmatic membranes of unsaturated fatty acids [37].

Our data agree with previous studies that confirm the antimicrobial potential of ozone. In our study, the maximum effective rate of ozone inhibition on *S. aureus* was 80.91% after 240 s of exposure. Similar results were reported for different strains of *Staphylococcus* (*Staphylococcus aureus*, 94.0%; *Staphylococcus epidermidis*, 88.6%; *Enterococcus faecalis*, 79.7%) [38]. In 2011, Wilczińska-Borawska et al. confirmed ozone’s bactericidal activity in bacterial strains most frequently isolated from the oral cavity. In addition, two gas application models on the infected medium were compared [39]. In agreement with our study, they found a statistically significant difference regarding inhibited bacterial growth on all media depending on the time of ozone action.

In 2000, Baysan et al. assessed the antimicrobial impact of ozone against *S. mutans* and *Streptococcus sobrinus* on primary root caries and observed an important inhibition effect for either 10 or 20 s after ozone introduction [40]. The killing rate was 78% after 6 s and 47% after twofold 18 s of exposure [40], but no completely eradicated strain was observed after applying ozone against *S. aureus*, *E. faecalis*, *Enterobacter cloacae*, and *C. albicans* [41]. Our results are in agreement with those obtained by Baysan et al. in 2000. The inhibition effect on *S. aureus* was proved at the first duration tested, meaning 5 s, with a BIR% value of 37.61% and a higher MIC value at 25 s. Against *S. mutans*, ozone significantly affected the bacterial load in dentin. Significant differences were found between the control and treated groups after 4 and 8 weeks after treatment, but no significant differences were found between 4 and 8 weeks [42].

The antibacterial effect of ozone on cariogenic bacterial species with and without the presence of saliva and a possible effect on the salivary proteins was studied by Johansson et al. in 2009 [43]. The results showed that in salt buffer, 92%, 73%, and 64% of the initial numbers of *A. naeslundii, S. mutans*, and *L. casei* were killed after 10 s of ozone exposure, and approximately 99.9% of the bacteria were dead after 60 s; however, in saliva, compared to the salt buffer, *S. mutans* and *L. casei* were less efficiently killed. In addition, saliva proteins were degraded by ozone after 60 s of exposure [43].

Huth KC et al., 2009 studied the influence of ozone aggregation conditions on the antimicrobial effect of treating different strains with not only either gaseous ozone or aqueous ozone, but also with 0.2%, 1%, and 2% chlorhexidine [44]. After 60 s of exposure, gaseous ozone at a concentration of 4 g m-3 was as effective as chlorhexidine. However, the highest concentration of aqueous ozone (20 μg/mL) proved even more effective than chlorhexidine in killing oral pathogens [44].

Although our experiments differed from those made by Huth et al. in 2009 regarding the time of exposure and ozone concentration, the results of this study agree with their results regarding the dose- and strain-dependence of ozone antimicrobial activity [44].

In another study, the effect of ozone on cultivatable microflora compared to chlorhexidine highlighted that the total reduction in the group without excavation was 7% after ozone treatment and 36% after chlorhexidine treatment. In comparison, 19% was the inhibition rate for ozone and 41% for chlorhexidine in the group with excavation [45]. The efficacy of ozone on cariogenic microorganisms (*S. mutans*, *S. salivarius*, *S. epidermidis*, *S. mitis*, *Lactobacillus*, and different kinds of *Staphylococcus*) exceeded the efficacy of 3% hydrogen peroxide and 0.2% chlorhexidine digluconate significantly [46].

In recent years, there has been an increasing emphasis on the association of ozonation with other natural preparations. The association of ozone with other natural preparations, such as essential oils, enhances its efficiency. The killing rates of ozonated oil against *S. aureus* were almost 100% [47]. In addition, ozonated water (4 mg/L) was discovered to be viable for treating gram-positive and gram-negative oral bacteria and oral *C. albicans* in plaque biofilms and helpful in controlling oral disease-causing microorganisms in teeth [48].

Montevecchi M. et al. in 2013 observed the antibacterial activity of ozonated sunflower oil against *Porphyromonas gingivalis* after 72 h of anaerobic incubation. Although ozonated oil showed a significantly greater zone of inhibition than 0.2% chlorhexidine against *S. aureus* and *P. gingivalis* [49], the evaluation of the antibacterial activity of ozonised olive oil against oral and periodontal pathogens compared with the other two gel agents, which were based on chlorhexidine, proved the moderate antiseptic potential of ozonated oil [22]. A previous study reported the effectiveness of ozonated oil on *Candida albicans*. Ozonated olive and sunflower oils demonstrated similar antimicrobial activity, with low MICs ranging from 0.53 to 0.2 µg/mL [23].

Other approaches have referred to the intensity and concentration of ozone applied [50]. Schneider H. in 2004 reported that ozone concentrations ranging between 300 and 800 ppm have bactericidal effects, with the maximal effect recorded at 525 ppm [51].

The efficiency of our method of using ozone mixed with MpEO as an antimicrobial agent is superior to methods reported by other authors who used only ozone. In the recent study of Rangel and al. in 2022, an inhibition rate of 17% against *P. aeruginosa* was reported after 60 s of exposure, and an inhibition effect of 99.99% against *S. aureus* was reported only after 40 min of exposure [52]. Our data highlighted the advantage of enhancing ozone with MpEO by showing the reduction of exposure time at 60 s with an inhibition rate of 65.71% against *P. aeruginosa* and 69.54% after 50 s against *S. aureus.*

In our study, the MIC values of MpEO varied. The MIC value of MpEO, reported by other authors, varied from 0.625 µg/mL against fungi such as *C. albicans* and *C. parapsilosis*, gram-positive bacteria such as *S. mutans*, *S. pyogenes* to 1.14–6.25 µg/mL for *E. coli* or 5 µg/mL for *P. aeruginosa* [34,35,53]. The MIC values obtained in our study for MpEO against *S. mutans*, *S. aureus*, *P. aeruginosa*, *E. coli*, and *C. albicans* were 2% lower than those reported by Sechi et al., who obtained MICs values against *S. aureus*, *Enterococcus faecalis*, *S. pyogenes*, *E. coli*, and *P. aeruginosa* that ranged between 1.18 and 9.5% [24].

Heydari in 2018 highlighted that MpEO possesses stronger antimicrobial activity with greater activity on the bacilli (*Bacillus subtilis* and *Bacillus cereus*) rather than on the cocci (*Staphylococcus aureus* and *Staphylococcus epidermidis*) [54]. Other authors reported only a moderate antibacterial effect against some tested bacteria [55].

The main chemical compounds of our MpEO are menthol (32.658%), menthyl acetate (23.643%), and menthone (10.322%). The MICs demonstrate that menthol is considerably toxic against *S. aureus* and the most toxic against *E. coli* [56]. The inhibition potential of these compounds against different bacteria and fungi is due to the toxic effects on their membrane structures and functions [57]. Monoterpenes have a lipophilic character responsible for the antimicrobial effects of menthol and menthone. The mechanism of action of menthol and menthone is a perturbation of the lipid fraction of a microorganism’s plasma membrane, resulting in membrane permeability alterations and leakage of intracellular materials [56].

The use of the ozone–MpEO mixture increased inhibition effects, as proved in this study, and this increase is based on the combination of the two effects and mechanisms of action of the materials as explained previously. The obtained results strengthen the applicability of these mixtures in medical and dental techniques.

## 4. Materials and Methods

### 4.1. The Obtaining and Characterisation of EOs 

The aerial parts of the investigated medicinal plant (Mp) were collected during the flowering period from the flora located in the outskirts of Timisoara, Romania (45°47′00.1″ N 21°12′37.2″ E) in 2020. A total of ~700 g fresh material was used. Additionally, a voucher specimen was botanically identified and deposited in the temperature-controlled herbarium (22–25 °C and 30–40% relative humidity) of the Faculty of Agriculture-Botany Department, University of Life Sciences “King Mihai I” Timisoara, Romania; the plant’s name (accepted name) was identified as *Mentha x piperita* L *(M. aquatica x M. spicata)* (Voucher Specimen Number Herbarium-Botany Department, VSNH.ULST-BD65).

The plant material was dried under natural conditions. The MpEO was extracted from 300 g of plant material using Clevenger-type equipment for 2 h, and the essential oil was separated via decantation and stored at 2–4 °C until used in GC/MS and microbiological analyses. The extraction yield of MpEO was calculated using the following formula:Yield (%) = [amount of EO (g)/amount of dry plant (g)] × 100(1)

The GC-MS characterisation of MpEO was done using Shimadzu QP 2010 Plus equipment (Shimadzu Corporation, Columbia, SC, USA), which was equipped with a capillary column AT WAX 30 m × 0.32 mm × 1 μm. The carrier gas used was helium, with a flow rate of 1 mL/min, and the injector and ion source temperatures were 250 °C and 220 °C, respectively. The gradient temperature was used for compound separation with an initial oven temperature of 40 °C maintained for 1 min; then, the temperature was raised to 210 °C at a rate of 5 °C/min and maintained for 5 min at this temperature. The sample injection volume was 1 μL, and a split ratio of 1:50 was used. Volatile compounds of MpEO were identified using the NIST5 Wiley 275 libraries database. The linear retention indexes (LRI) were calculated using n-alkane C8–C27 standards using the same experimental conditions. The results are presented as percentages from total compounds [24]. The chromatogram of the chemical compounds is presented in the Appendix A.

### 4.2. Microbiological Assay

#### 4.2.1. Microbial Strains

All of the strains used belong to the culture collection of the Microbiology Laboratory of the Interdisciplinary Research Platform within the University of Life Sciences “King Michael I” from Timișoara. The used microbial strains were: *Streptococcusmutans* (ATCC 19615), *Staphylococcus aureus* (ATCC 25923), *Escherichia coli* (ATCC 25922), *Pseudomonas aeruginosa* (ATCC 27853), and *Candida albicans* (ATCC 10231). All ATCC strains are maintained at −50 °C.

#### 4.2.2. Microbial Culture Preparation and Essential Oil Efficacy Assessment

Antimicrobial activity was tested according to the method described by Hulea et al. [58] and Obistioiu et al. [59], and the CLSI method was used for microdilution for the antimicrobial susceptibility test [60]. Different concentrations of MpEO ranging from 2 µL–32 µL were tested using the broth microdilution method. Considering that the quantity of oil tested was immersed in 100 µL of Brain Heart Infusion Broth, the concentration reached is expressed as a percentage (2–32%). Subsequently, the minimum inhibitory concentration (MIC) was determined. The MIC is defined as the lowest compound concentration that yields no visible microorganism growth, and it is based on mass loss by measurement of optical density (OD) by using the spectrophotometric method at 540 nm. For ozone, MIC was determined as the lowest time expressed in seconds of the ozone or enhanced ozone exposure that inhibited the visible growth of cells. The values obtained are expressed as OD reading values (values presented as mean ± standard deviation in Appendix A). The simple strain multiplied in BHI was used as a negative control. The control group for each strain was represented by the ATCC strain multiplied in BHI after 24 h without any interference. The mean OD values reached for each experiment were used in Formulas (2) and (3) and were considered to be 100% BGR and 0% BIR for each strain.

Two indicators were calculated, MGR/BGR and MIR/BIR, to interpret the values obtained by using the following formulas:(2)$$\mathrm{MGR}/\mathrm{BGR}=\frac{OD_{sample}}{OD_{negativecontrol}}\times 100\left(\%\right)$$
MIR/BIR = 100 − MGR/BGR (%)(3)
where:

*OD sample*—optical density at 540 nm as the mean value of triplicate samples in the presence of the selected fungi/bacteria;

*OD negative control*—optical density at 540 nm as the mean value of triplicate readings for the selected fungi/bacteria in BHI.

#### 4.2.3. Ozonation Procedure

Various authors researched the effect of ozone in either clinical or microbiological studies, all stating as the conclusion that the duration of action can be an important consideration in ozone’s antibacterial effect. Noites et al. [61] used gaseous ozone applied with an ozone generator for 24, 60, 120, and 180 s and assessed the antimicrobial impact of ozone against *S. mutans* and *Streptococcus sobrinus* for either 10 or 20 s, whereas Baysan et al. [40,62] assessed that an ozone application of 10–20 s eliminated more than 99% of the microorganisms found in dental caries and associated biofilms, and a 40 s treatment time covered all eventualities. Starting from these conclusions, we selected a wider range of time periods to investigate the efficacy of ozone on the microorganisms tested to ensure we obtained the necessary MIC.

The ATCC strains were successively subjected to ozonation with gaseous O_3_ produced with HealOzone X4 (KaVo Dental & Co., Biberach, Germany), which released O_3_ at a fixed concentration of 2100 ppm with a flow rate of 615 cm^2^/min for 5″ (s), 10″, 15″, 20″, 25″, 30″, 35″, 40″, 45″, 50″, 55″, 60″, 120″, 180″, and 240″. Each period selected in our research was measured according to the times suggested by the manufacturer and in research articles.

The ozonation device comprises an air filter, vacuum pump, an ozone generator, a handpiece fitted with a sealing silicone cup, and a flexible hose. Once the procedure was started, the hose end was submersed into the ATCC strain within the 96-microdilution plate, and therefore, no O_3_ was lost from the dose injected in the BHI broth. According to the specifications, the device produced ozone at a concentration of 32 g/m^3^ at an exposure of 60 s.

This analysis method was also used for broth microdilution, and all of the previous steps presented before in the microbiological methodology (Section 4.2.1 and Section 4.2.2) were followed once the ozonation procedure was finished.

### 4.3. Statistical Analysis

Differences between means were analysed with a one-way ANOVA, followed by multiple comparisons using a *t*-test (two-sample assuming equal variances) using GraphPad Prism 8 (version 8.0.2; GraphPad Software Inc., La Jolla, CA, USA). Differences were considered significant when *p*-values < 0.05. Time–inhibition relationships were established with the Microsoft Excel 365 software.

## 5. Conclusions

The results underline that ozone in combination with MpEO exerts synergistic effects of potentiating antimicrobial activity. The antibacterial effects obtained depend on contact time and the strain type. The effect of ozone on the following tested strains at maximum efficiency was observed after 55 s of single ozone exposure, listed in descending order of effect strength: *S. aureus* > *P. aeruginosa* > *E. coli* > *C. albicans* > *S. mutans.* Exposure to the ozone–MpEO mixture decreased contact time with the bacteria, which is of practical and economic importance. For ozone with the addition of 2% MpEO, the maximum efficacy was recorded at 5 s in the following strains, listed in descending order of effect strength: *C. albicans* > *E. coli* > *P. aeruginosa* > *S. aureus* > *S. mutans*.

The results demonstrate that the combination of ozone with MpEO leads to an increase in efficiency and a decrease in exposure time. All bacterial strains were inhibited, with BIR% values ranging from 36.59% to 74.06%, and the exposure time was reduced from 120 s to an optimal 55 s.

In the search for alternative antimicrobial solutions to synthetic antibiotics, the use of ozone in combination with essential oils can represent innovative solutions with applications in medicine and dentistry.

## Figures and Tables

**Figure 1 molecules-28-02032-f001:**
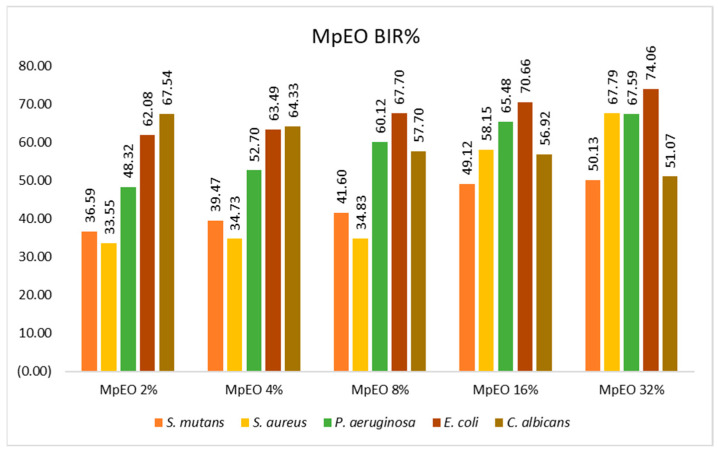
MpEO antimicrobial activity on the tested strains at different concentrations expressed as BIR%.

**Figure 2 molecules-28-02032-f002:**
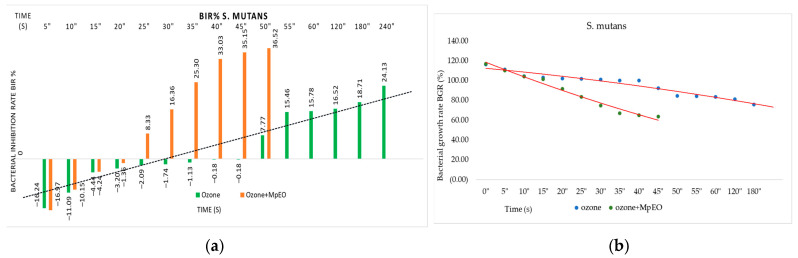
Ozone and ozone+MpEO antimicrobial activity on *S. mutans* at different time exposures expressed in seconds (″). (**a**) Results expressed as BIR%; (**b**) results expressed as BGR%.

**Figure 3 molecules-28-02032-f003:**
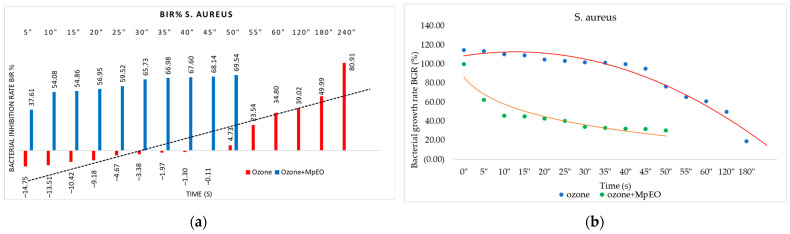
Ozone and ozone+MpEO antimicrobial activity on *S. aureus* at different time exposures expressed in seconds (″). (**a**) Results expressed as BIR%; (**b**) results expressed as BGR%.

**Figure 4 molecules-28-02032-f004:**
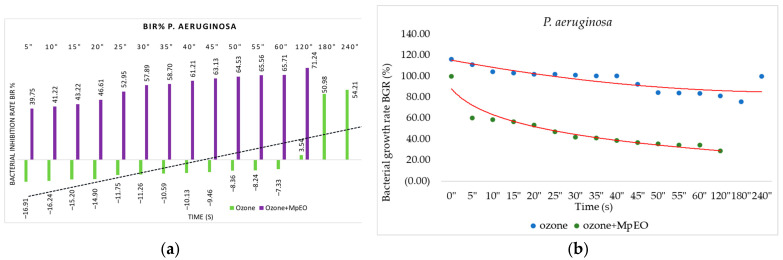
Ozone and ozone+MpEO antimicrobial activity on *P. aeruginosa* at different time exposures expressed in seconds (″). (**a**) Results expressed as BIR%; (**b**) results expressed as BGR%.

**Figure 5 molecules-28-02032-f005:**
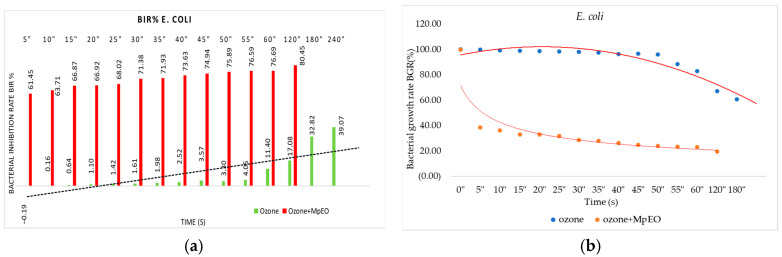
Ozone and ozone+MpEO antimicrobial activity on *E. coli* at different time exposures expressed in seconds (″). (**a**) Results expressed as BIR%; (**b**) results expressed as BGR%.

**Figure 6 molecules-28-02032-f006:**
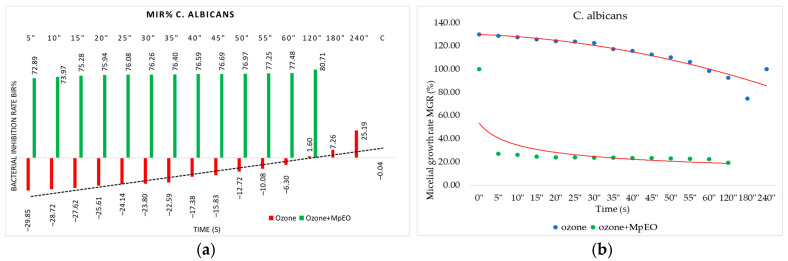
Ozone and ozone+MpEO antimicrobial activity tested on *C. albicans* at different time exposures expressed in seconds (″). (**a**) Results expressed as MIR%; (**b**) results expressed as BGR%.

**Figure 7 molecules-28-02032-f007:**
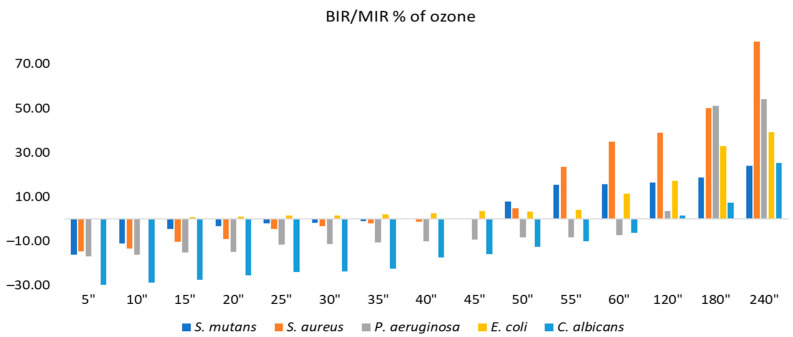
Antimicrobial activity (expressed as BIR/MIR%) of ozone tested on ATCC strains at different exposure times represented in seconds (″).

**Figure 8 molecules-28-02032-f008:**
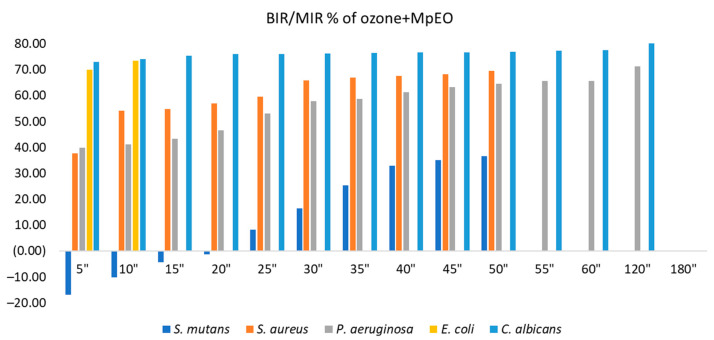
Antimicrobial activity (expressed as BIR/MIR%) of ozone+MpEO tested on ATCC strains at different exposure times expressed in seconds (″).

**Table 1 molecules-28-02032-t001:** Chemical composition of MpEO.

Compounds	Type	LRI ^a^	% of Total Compounds
Linalool	MO	1533	0.752
α-pinene	MH	1021	1.476
β-pinene	MH	1106	1.179
Sabinene	MH	1135	0.508
Limonene	MH	1196	3.137
β-trans-ocimene	MH	1199	0.233
γ-terpinene	MH	1202	0.399
Eucalyptol	MO	1204	5.639
*p*-cymene	MH	1284	0.454
Menthofurane	MO	1474	2.960
Linalol acetate	MO	1541	0.558
Menthyl-acetate iso	MO	1546	0.375
Menthyl acetateone racemic	MO	1548	10.322
*p*-menthan-3-one	MO	1552	23.643
Menthol, acetate, iso-	MO	1562	1.882
Isomenthone	MO	1582	4.046
Terpinen-4-ol	MO	1593	1.191
Menthol	MO	1634	32.648
Germacrene D	SH	1708	5.410
γ-Elemene	SH	1717	0.244
*p*-menth-1-en-8-ol	MO	1724	0.433
Pulegone	MO	1730	1.636
Piperitone	MO	1750	0.521
Caryophyllene oxide	SO	1889	0.355
Total of compounds			100.000
Monoterpene hydrocarbonates	MH		7.386
Monoterpene oxygenate	MO		86.606
Sesquiterpene hydrocarbonates	SH		5.653
Sesquiterpene oxygenates	SO		0.355

^a^ Linear retention indices (LRI) calculated according to n-alkanes (C8–C27) for AT WAX 30 m × 0.32 mm × 1 μm column [24].

**Table 2 molecules-28-02032-t002:** The MIC values of ozone (a), ozone+MpEO (b), and MpEO (c) on ATTC strains at different exposure times. The red background colour highlights the time duration or concentration where the MIC was determined for each strain. The effect was maintained together with an increase in the period/concentration. The ozonations with positive inhibitory responses are marked with grey background.

MIC Values of Ozone						MIC Values of Ozone+MpEO					
*S.* *mutans*	*S.* *aureus*	*P.* *aeruginosa*		*E.* *coli*	*C.* *albicans*	*S.* *mutans*	*S.* *aureus*	*P.* *aeruginosa*		*E.* *coli*	*C.* *albicans*
5″	5″	5″		5″	5″	5″	5″	5″		5″	5″
10″	10″	10″		10″	10″	10″	10″	10″		10″	10″
15″	15″	15″		15″	15″	15″	15″	15″		15″	15″
20″	20″	20″		20″	20″	20″	20″	20″		20″	20″
25″	25″	25″		25″	25″	25″	25″	25″		25″	25″
30″	30″	30″		30″	30″	30″	30″	30″		30″	30″
35″	35″	35″		35″	35″	35″	35″	35″		35″	35″
40″	40″	40″		40″	40″	40″	40″	40″		40″	40″
45″	45″	45″		45″	45″	45″	45″	45″		45″	45″
50″	50″	50″		50″	50″	50″	50″	50″		50″	50″
55″	55″	55″		55″	55″	55″	55″	55″		55″	55″
60″	60″	60″		60″	60″	60″	60″	60″		60″	60″
120″	120″	120″		120″	120″	120″	120″	120″		120″	120″
180″	180″	180″		180″	180″	180″	180″	180″		180″	180″
240″	240″	240″		240″	240″	240″	240″	240″		240″	240″
		** *S. mutans* **		** *S. aureus* **		** *P. aeruginosa* **		** *E. coli* **		** *C. albicans* **	
MpEO%		2		2		2		2		2	
MpEO%		4		4		4		4		4	
MpEO%		8		8		8		8		8	
MpEO%		16		16		16		16		16	
MpEO%		32		32		32		32		32	

## Data Availability

Not applicable.

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
