# Peer review of "Enhancing the Antimicrobial Effect of Ozone with Mentha piperita Essential Oil"

_molecules, 2023, doi:10.3390/molecules28052032_

Round 1
Reviewer 1 Report (New Reviewer)
Please my comments below:
· Line 79: second part of latin names of microorganism should be written in small letter.
· Table 1: the alpha and beta should be written as greek letters.
· Why do the literature RI values not agree with those obtained in this experiment?
· Why is the analysis of compounds in the essential oil limited to only a few components?
· How exactly was the oil obtained? Please provide all data for the methodology.
· How was the composition of the oil tested? Please specify equipment, column, conditions, etc.
· Who confirmed the botanical identity of the plant?
Author Response
Dear Reviewer,
We would like to address all our thanks and gratitude for the constructive observations, corrections and recommendations.
Comments and Suggestions for Authors
Comment : Line 79: second part of latin names of microorganism should be written in small letter.
Answer: The corrections were done
Comment: Table 1: the alpha and beta should be written as greek letters.
Answer:The corrections were done
Comment: Why do the literature RI values not agree with those obtained in this experiment?
Answer: The LRI provided from the literature data are in similar experimental conditions but not identical to those in the current work, and as a result some differences appear between the LRI determined in our experimental conditions and those found in the literature.
Comment: Why is the analysis of compounds in the essential oil limited to only a few components?
Answer: In the table 1, the compounds identified in a percentage higher than 1% were included, considering that those below this value are not useful for the present study. At the reviewer's suggestion, they were also presented in the table.
Comment: How exactly was the oil obtained? Please provide all data for the methodology.
How was the composition of the oil tested? Please specify equipment, column, conditions, etc.
Answer: In subsection 4.1. of the materials and method, the method of EO extraction and the analysis of the main chemical compounds by GC-MS are presented.
Comment: Who confirmed the botanical identity of the plant?
Answer: The section 4.1. was completed with the following text: „ The aerial parts of the investigated medicinal plant (Mp) were collected during the flowering period from the flora located in the outskirts of Timisoara, Romania (45°47'00.1"N 21°12'37.2"E) in 2020. ~700 g fresh material was used. Additionally, a voucher specimen was botanically identified and deposited in the temperature-controlled herbarium (22–25°C and 30–40% relative humidity) of the Faculty of Agriculture - Botany Department, University of Life Sciences "King Mihai I" Timisoara, Romania, plants name (accepted name) identified as Mentha x piperita L. (M. aquatica x M. spicata) (Voucher Specimen Number Herbarium - Botany Department, VSNH.ULST-BD65). „.
Once again, we would like to thank the reviewer for your appreciations, corrections and recommendations which contributed to the significant improvement of the paper.
Reviewer 2 Report (New Reviewer)
The manuscript evaluates use of Mentha piperita essential oil (MpEO) as an enhancement agent for the antimicrobial potential of ozone against Gram-positive, Gram-negative bacteria and fungi at different exposure times. The antimicrobial activity of essential oil and ozone is already known. The application of many essential oils having antimicrobial properties and also plant extracts with and without oxidizing agent is already well reported in recent years for eliminating various bacteria (Mane et al., 2020 https://doi.org/10.1016/j.ultsonch.2020.105272; Mane et al., 2021 https://doi.org/10.1016/j.jwpe.2021.102280 )
The authors need to not just refer to these, but should also comment in the light of the recent developments to highlight the importance/ advancements / new addition to the literature.
The article may be improved using the following suggestions:
1. Experiment procedure needs to be provided with sufficient details, e.g. addition of the oil? Combination of oil/ ozone; use of different MpEO concentration etc.
2. Please clarify use of 2 % concentration of MpEo as reported use of MIC ranges in µL (2µL - 32 µL).
3. It is essential that the results be elaborately analysed and compared with the recent developments. Plausible mechanism for various bacteria and fungi be discussed, e.g. for the different behavior of BIR to different types of bacteria; for enhancements due to ozone, concentration effect and interpretation.
4. Introduction needs to also provide information on other methods, advantages/ disadvantages and highlight the objectives of the research. Restructure Sec. 2.2 for improved presentation of the work.
5. Need to improve the quality of the figures, for better readability. Please correct figures titles (e.g. Fig. 1, Fig. 6, Fig. 9). In some places, Table is referred as Figure. Table-2 may not be needed. Need to provide uncertainty analysis.
Author Response
Dear Reviewer,
We would like to address all our thanks and gratitude for the constructive observations, corrections and recommendations.
Comments and Suggestions for Authors
The manuscript evaluates use of Mentha piperita essential oil (MpEO) as an enhancement agent for the antimicrobial potential of ozone against Gram-positive, Gram-negative bacteria and fungi at different exposure times. The antimicrobial activity of essential oil and ozone is already known. The application of many essential oils having antimicrobial properties and also plant extracts with and without oxidizing agent is already well reported in recent years for eliminating various bacteria (Mane et al., 2020 https://doi.org/10.1016/j.ultsonch.2020.105272; Mane et al., 2021 https://doi.org/10.1016/j.jwpe.2021.102280)
The authors need to not just refer to these, but should also comment in the light of the recent developments to highlight the importance/ advancements / new addition to the literature.
Answer: Thank you for the relevant comments, the literature study in the introduction has been improved and new bibliographic references have been added. The recent developments and the importance of research to the literature studies were presented.
The article may be improved using the following suggestions:
Comment : 1. Experiment procedure needs to be provided with sufficient details, e.g. addition of the oil? Combination of oil/ ozone; use of different MpEO concentration etc.
Answer: The methodology part was modified and completed following your suggestions.
Comment : 2. Please clarify use of 2 % concentration of MpEo as reported use of MIC ranges in µL (2µL - 32 µL).
Answer: The explanation is inserted in Sec. 4.2.2. Different concentrations of MpEO were tested, ranging from 2µL - 32 µL, using the broth microdilution method. Considering the fact that the quantity tested of oil was done using 100µL of Brain Heart Infusion Broth, the concentration reached will be expressed as a percentage (2-32%).
Comment : 3. It is essential that the results be elaborately analysed and compared with the recent developments. Plausible mechanism for various bacteria and fungi be discussed, e.g. for the different behavior of BIR to different types of bacteria; for enhancements due to ozone, concentration effect and interpretation.
Answer:In the section 3.2. our results are compared with the recent developments. Also, the mechanism of bacteria inhibition by ozone and MpEO are discussed. The enhancements of antimicrobial activity of ozone due concentration are presented and interpreted. Also, the antimicrobial effect of MpEO and they chemical components are explained.
Comment : 4. Introduction needs to also provide information on other methods, advantages/ disadvantages and highlight the objectives of the research. Restructure Sec. 2.2 for improved presentation of the work.
Answer: Introduction was upgraded with information about other methods used regarding ozon application including hybrid hydrodynamic cavitation process and their advantages are presented. The objectives of the research were presented. Section 2.2. and 2.3. were improved.
- Need to improve the quality of the figures, for better readability. Please correct figures titles (e.g. Fig. 1, Fig. 6, Fig. 9). In some places, Table is referred as Figure. Table-2 may not be needed. Need to provide uncertainty analysis.
Answer: The modifications were made concerning the tables and the figures suggested. Table 2 was removed and submitted as a supplementary file.
Once again, we would like to thank the reviewer for your appreciations, corrections and recommendations which contributed to the significant improvement of the paper.
Reviewer 3 Report (New Reviewer)
This manuscript describes Enhancing the Antimicrobial Effect of Ozone with Mentha 2 Piperita Essential Oil
The objective of this work is to obtain and analyse Mentha piperita essential oil (MpEO) to be used as an enhancement agent for the antimicrobial potential of ozone against Gram-positive, Gram-negative bacteria and fungi at different exposure times and gaining a time-dose relationship as well as a time-effect correlation.. Therefore, in my opinion, the paper would merit acceptance to molecules after addressing the following specific items. Major issues:
1) The introduction section does not sufficiently state plainly and unambiguously what are the hypotheses and objectives of this work, especially regarding the targeted reactivity. It is important to clarify this point and better highlight what is the originality of the present study both in introduction and conclusion. Furthermore, the analysis of literature provided in the introduction section is lacking major contributions closely related to this work. The authors should describe the results from these studies in the introduction section, and should be compared more carefully with those of state of the art systems in the discussion section.
2) It has been brought to our attention that there is a significant amount of textual overlap in your manuscript from the previously published work which is totally unethical and not suitable to be published. You are requested to rewrite these paragraphs .
3) references should be update, there’s many new work in this connexion
4) The antimicrobial activity tests were not determined by which methods? well, method or disc diffusion? How bacteria counts were determined?
1) It is not clear how the initial bacterial count (CFU/ml) was adjusted from stock bacterial cultures
6) How and by what control groups were created?
7) The authors must include in the methodology the name and the origin of the microorganism tested. For example - Staphylococcus aureus NTCTC 7447 (National Collection of Type Cultures) - this information is misplaced in the Results section, and should appear in Experimental Section .
Some information in the Determination of Minimum Inhibitory Concentration (MIC) of Bioactive Metabolite needs to be explained:
Disc diffusion and micro broth dilution methods against standard antibiotic discs; In general, the following international standards are applied, Although the method is given correctly, standards are not specified
http://www.eucast.org/
https://clsi.org/
There is no references
8) There is No NMR and MS spectra are given in the supporting information
9) The point-wise summary of analytical techniques is not a proper discussion of results
10) Authors should explain how combination of ozone with MpEO, exerts synergistic effects of potentiating antimicrobial activity
11) Authors make the extraction of MpEO from the aerial parts of Mentha piperita, what about the characterization of MpEO ?
12) the animicrobial is related exactly to which compound, authors know the strucutre ?
Author Response
Dear Reviewer,
We would like to address all our thanks and gratitude for the constructive observations, corrections and recommendations.
Comments and Suggestions for Authors
The objective of this work is to obtain and analyse Mentha piperita essential oil (MpEO) to be used as an enhancement agent for the antimicrobial potential of ozone against Gram-positive, Gram-negative bacteria and fungi at different exposure times and gaining a time-dose relationship as well as a time-effect correlation.. Therefore, in my opinion, the paper would merit acceptance to molecules after addressing the following specific items.
Major issues:
Comment: 1) The introduction section does not sufficiently state plainly and unambiguously what are the hypotheses and objectives of this work, especially regarding the targeted reactivity. It is important to clarify this point and better highlight what is the originality of the present study both in introduction and conclusion. Furthermore, the analysis of literature provided in the introduction section is lacking major contributions closely related to this work. The authors should describe the results from these studies in the introduction section, and should be compared more carefully with those of state of the art systems in the discussion section
Answer: Thank you for the relevant comments, the literature study in the introduction has been improved and new bibliographic references have been added. The recent developments and the importance of research to the literature studies were presented. The originality of this study is presented in the introduction and in conclusion. The results from the literature studies were added in the introduction section. In the discussion section, the obtained results were compared with those of state of the art.
Comment: 2) It has been brought to our attention that there is a significant amount of textual overlap in your manuscript from the previously published work which is totally unethical and not suitable to be published. You are requested to rewrite these paragraphs .
Answer: The methodology sections were modified not to have overlaps with our previous works.
Comment: 3) references should be update, there’s many new work in this connexion
Answer: The references were updated and new titles were added in the field of research.
Comment: 4) The antimicrobial activity tests were not determined by which methods? well, method or disc diffusion? How bacteria counts were determined?
Answer: The method used was the broth microdilution method, and the bacteria count was determined by using the McFarland standard The cultures were diluted at an optical density (OD) of 0.5 McFarland standard (1.5 × 108UFC×mL) using BHI broth and a McFarland densitometer Grand-Bio ( Fisher Scientific, Loughborough, UK). This is presented in section 4.2.2 as cited reference.
Comment: 5) It is not clear how the initial bacterial count (CFU/ml) was adjusted from stock bacterial cultures
Answer: Using the stock sollution, dillutions were made to reach a 0.5 McFarland standard (1.5 × 108UFC×mL) using BHI broth and a McFarland densitometer Grand-Bio ( Fisher Scientific, Loughborough, UK).
Comment: 6) How and by what control groups were created?
Answer: The control group for each strain was represented by the ATCC strain multiplied in BHI after 24 hours without any interference. The OD value reached were used in formulas (1) and (2) and were considered as 100% BGR and 0% BIR for each strain.
Comment: 7) The authors must include in the methodology the name and the origin of the microorganism tested. For example - Staphylococcus aureus NTCTC 7447 (National Collection of Type Cultures) - this information is misplaced in the Results section, and should appear in Experimental Section.
Answer: The corrections were made within the Methodology section.
Comment: Some information in the Determination of Minimum Inhibitory Concentration (MIC) of Bioactive Metabolite needs to be explained:
Disc diffusion and micro broth dilution methods against standard antibiotic discs; In general, the following international standards are applied, Although the method is given correctly, standards are not specified
http://www.eucast.org/
https://clsi.org/
There is no references
Answer: Thank you, the references were inserted in the manuscript.
Comment: 8) There is No NMR and MS spectra are given in the supporting information
Answer: The chromatogram of MpEO was added as Supplementary material
Comment: 9) The point-wise summary of analytical techniques is not a proper discussion of results
Answer: The analytical technique used for the characterization of MpEO is presented in the section 4.1. and refers to GC-MS. The results regarding the chemical composition of MpEO, including retention times, LRI and percentage composition are presented in table 1 and discussed in relation to the results from the literature in the section 3.1. The microbiological analysis aimed to determine the optical density (OD) whose values are presented in the supplementary material S1, and the interpretations were made based on the BIR/MIR, BGR/MGR values determined according to relations 2-3 in the material and method section 4.2.
Comment: 10) Authors should explain how combination of ozone with MpEO, exerts synergistic effects of potentiating antimicrobial activity
Answer: All the results obtained using the OD values and, subsequently, the BIR values demonstrate the synergistic effect of ozone and MpEO mixture. This affirmation is sustained by the fact that the inhibitory rates obtained in the mixture have increased compared to the inhibitory rates obtained in the case of ozone and MpEO tested as single compounds. This fact is possibly a result of a dual effect: the oxidation reaction of the bacterial wall produced by the ozone, potentiated with the MpEO mechanism of weakening the cell wall of resistant bacteria, thereby damaging or killing the cells.
Comment: 11) Authors make the extraction of MpEO from the aerial parts of Mentha piperita, what about the characterization of MpEO ?
Answer: The method of sampling the plant material of Mentha piperita, its processing and the extraction of the MpEO essential oil are presented in the section 4.1.The plant material was represented by the aerial parts from which the essential oil was obtained using the Clevenger equipment and characterized using GC-MS.
Comment: 12) the animicrobial is related exactly to which compound, authors know the strucutre ?
Answer: In the penultimate paragraph of subsection 3.4, the main components of MpEO and the antimicrobial mode of action are presented.
Once again, we would like to thank the reviewer for your appreciations, corrections and recommendations which contributed to the significant improvement of the paper.
Round 2
Reviewer 1 Report (New Reviewer)
Please standardize the literature RI according to the NIST database. Please consider the analysis conditions and the type of column. At present, there is no certainty about the identification of the compounds in the tested oil.
Author Response
Dear Reviewer,
We would like to address all our thanks and gratitude for the constructive observations, corrections and recommendations.
Comments and Suggestions for Authors
Please standardize the literature RI according to the NIST database. Please consider the analysis conditions and the type of column. At present, there is no certainty about the identification of the compounds in the tested oil.
Answer: The LRI values reported in the column 3, table 1, were taken from the Nist database (https://webbook.nist.gov/). For each component the appropriate values were chosen depending on the type of chromatographic column and the separation conditions. The bibliographic reference chosen, for each compound, corresponds to the author who reported the LRI values in the separation conditions similar to those experienced in the present study.
For example, for alpha pinene from Nist database https://webbook.nist.gov/cgi/cbook.cgi?ID=C80568&Units=SI&Mask=2000#Gas-Chrom it was chosen the LRI value 1001 reported by Giuseppe, Manuela, et al., 2005 separated on DB-Wax column and separation conditions: 30. m/0.25 mm/0.25 μm, He; Program: 35C(5min) => 2C/min => 173C(1min) => 15C/min => 210C(5min), reference 26 in the paper. This was the procedure for each of the experimentally separated compounds.
The experimental compounds were identified using the NIST 5 Wiley 275 libraries database. The Linear retention indexes (LRI) were calculated using n-alkanes C8-C27 standards in the same experimental conditions. The results were presented as percentages from total compounds. The method of performing the analysis and the experimental conditions are presented in chapter 4.1.
Once again, we would like to thank the reviewer for your appreciations, corrections and recommendations which contributed to the significant improvement of the paper.
Reviewer 2 Report (New Reviewer)
1. The statement, “Thus, this study presents novel elements …” may be avoided since ozone and such oxidizing agents are already reported. Please reword the sentence.
2. Ozonation procedure is still not very clear. Please provide basis for the conditions or reference.
3. In results, please also clearly highlight efficacy of the method over literature reports.
4. The conclusion, “In the search for alternative antimicrobial solutions … ozone in combination with essential oils can represent a novative and viable solution with applications in medicine and dentistry.”- Viability is not proved. Please reword appropriately.
5. Please highlight the results in conclusion by providing quantitative values, “The results demonstrate that the association of ozone with MpEO leads to an increase in efficiency and a decrease in exposure time.”
1. Error bars may be provided in the figures. Check Fig. 6 Chart title (BIR %) and rectify.
2. The quality of the figures is still not up to the mark. Please improve.
Author Response
Dear Reviewer,
We would like to address all our thanks and gratitude for the constructive observations, corrections and recommendations.
Comments and Suggestions for Authors
- Comment: The statement, “Thus, this study presents novel elements …” may be avoided since ozone and such oxidizing agents are already reported. Please reword the sentence.
Answer: The sentence was deleted
- Comment: Ozonation procedure is still not very clear. Please provide basis for the conditions or reference.
Answer Various authors researched the effect of ozone in either clinical studies or microbiology, and all concluded that the duration of action can be an important consideration in ozone antibacterial effect. Noites et al., used gaseous ozone applied with an ozone generator for 24, 60, 120, and 180 seconds and assessed the antimicrobial impact of ozone against S. mutans and Streptococcus sobrinus for either 10 or 20 seconds, while Baysan et al., 2004 assessed that an ozone application of 10 – 20 second has been reported to eliminate more than 99% of the microorganisms found in the dental caries and associated biofilms – and a 40 second treatment time covers all eventualities. Starting from these conclusions, we selected a wider range of seconds to investigate the efficacy of ozone on the microorganisms tested to ensure obtaining the necessary MIC.
- Comment: In results, please also clearly highlight efficacy of the method over literature reports.
Answer: Examples of the efficiency of the method compared to the data reported in the literature were added, as follows:
``The MIC values obtained in our study for MpEO was 2% against S. mutans, S. aureus, P. aeruginosa, E. coli, C. albicans, value lower than those reported by Sechi et al., thatwhich reported who obtained MICs values against S. aureus, Enterococcus faecalis, S. pyogenes, E. coli and P. aeruginosa, between 1.18 to 9.5% mg/ml [24]``.
And:
``The efficiency of our method of using ozone mixed with MpEO as an antimicrobial agent is superior to that reported by other authors who used only ozone. In the recent study of Rangel and al., 2022 it was reported an inhibition rate of 17% against P.aeruginosa after 60 seconds of exposure and against S.aureus an inhibition effect of 99.99% was reported only after 40 minutes of exposure [Rangel and al., 2022]. Our data highlighted the advantage of the association of MpEO justified by the reducing of exposure time at 60 seconds with an inhibition rate of 65.71% against P.aeruginosa and 69.54% after 50 seconds against S.aureus. ``
- Comment: The conclusion, “In the search for alternative antimicrobial solutions … ozone in combination with essential oils can represent a novative and viable solution with applications in medicine and dentistry.”- Viability is not proved. Please reword appropriately.
Answer: The sentence was reformulated.
- Comment:Please highlight the results in conclusion by providing quantitative values, “The results demonstrate that the association of ozone with MpEO leads to an increase in efficiency and a decrease in exposure time.”
Answer: The explanation was added in the conclusions: „All bacterial strains reacted through inhibition, for each the trend being a positive one, with BIR% values ranging from 36.59% to 74.06% and the exposure time was reduced from 120 seconds to an optimal 55 seconds``
- Comment: Error bars may be provided in the figures. Check Fig. 6 Chart title (BIR %) and rectify.
Answer: Statistical calculation and error bars were applied for the determined OD values (values not presented in the text, but insered as Supplementary material. BIR(%) was calculated for the mean value of OD, so no error bars are applied for these values. The explanation was added in the text at 4.2.2. section from methods. The title and the figure 6 were upgraded.
- Comment:The quality of the figures is still not up to the mark. Please improve.
Answer: The figures were improved
Once again, we would like to thank the reviewer for your appreciations, corrections and recommendations which contributed to the significant improvement of the paper.
Reviewer 3 Report (New Reviewer)
The authors have revised their manuscript by considering the reviewers' comments but it still to update some reference and rewrite some paragraphs to ovoid plagiarism
Author Response
Dear Reviewer,
We would like to address all our thanks and gratitude for the constructive observations, corrections and recommendations.
Comments and Suggestions for Authors
The authors have revised their manuscript by considering the reviewers' comments but it still to update some reference and rewrite some paragraphs to ovoid plagiarism
Answer: The references were updated and the manuscript has been checked for plagiarism. The percentage found was acceptable, however, corrections were made where necessary.
Once again, we would like to thank the reviewer for your appreciations, corrections and recommendations which contributed to the significant improvement of the paper.
This manuscript is a resubmission of an earlier submission. The following is a list of the peer review reports and author responses from that submission.
Round 1
Reviewer 1 Report
The author reported enhancing the antimicrobial effect of ozone with mentha piperita essential oil. The research can attracted the reader’s interest in this field. However, some aspects need to be concerned before considering the acceptance.
1. Why not show the antimicrobial activity of essential oil in absence of ozone for the aims of comparison, although author claimed that the previous research of MpEO showed low values of MIC at about 2% for all the strains tested.
2. In antimicrobial experiments, the methods to apply ozone for antimicrobial activity should be detailed, for example, equipment, concentration, and so on.
Author Response
Dear Reviewer,
We would like to address all our thanks and gratitude for the constructive observations, corrections and recommendations.
Comments and Suggestions for Authors
The author reported enhancing the antimicrobial effect of ozone with mentha piperita essential oil. The research can attracted the reader’s interest in this field. However, some aspects need to be concerned before considering the acceptance.
- Why not show the antimicrobial activity of essential oil in absence of ozone for the aims of comparison, although author claimed that the previous research of MpEO showed low values of MIC at about 2% for all the strains tested.
Answer: Based on the reviewers’ recommendations, the authors added a figure with data on the antimicrobial activity of MpEO (figure 8c) and interpretations.
- In antimicrobial experiments, the methods to apply ozone for antimicrobial activity should be detailed, for example, equipment, concentration, and so on.
Answer: Elements regarding the working method have been added in paragraph 4.2.3.
Once again, we would like to thank the reviewer for your appreciation, corrections and recommendations, which contributed to the significant improvement of the paper.
Reviewer 2 Report
The antimicrobial properties of peppermint oil and ozone are already well known and described in the literature, which means that the presented work is characterized by a lack of novelty. Therefore, I believe that this manuscript is not appropriate for publication in Molecules.
Author Response
Thank you for your input, the reviewers corrections and recommendations contributed to the significant improvement of the paper.
Reviewer 3 Report
The manuscript is fine, however, recent literature needs to be cited. Conclusion needs to be improved significantly. The authors have used GC-MS for identification of compounds. The spectra may be provided in supplementary material.
Author Response
Dear Reviewer,
We would like to address all our thanks and gratitude for the constructive observations, corrections and recommendations.
Comments and Suggestions for Authors
- The manuscript is fine, however, recent literature needs to be cited.
Answer: Based on the reviewers’ recommendations, the author added new literature data regarding chemical composition and antimicrobial activity of Mentha piperita essential oil and ozone.
- Conclusion needs to be improved significantly.
Answer: The conclusions were improved.
- The authors have used GC-MS for identification of compounds. The spectra may be provided in supplementary material.
Answer: The GC-MS spectra were added as supplementary materials.
Once again, we would like to thank the reviewer for your appreciation, corrections and recommendations, which contributed to the significant improvement of the paper.
Round 2
Reviewer 1 Report
The manuscript has been revised according to reviewer's comment. It can be considered for acceptance.
Reviewer 2 Report
-
Reviewer 3 Report
The authors have sufficiently revised the manuscript and I am satisfied with the revision. The manuscript can now be accepted for publication.